# Coronarin D Induces Apoptotic Cell Death and Cell Cycle Arrest in Human Glioblastoma Cell Line

**DOI:** 10.3390/molecules24244498

**Published:** 2019-12-09

**Authors:** Yollanda E. M. Franco, Marcia Y. Okubo, Adriana D. Torre, Paula P. Paiva, Marcela N. Rosa, Viviane A. O. Silva, Rui M. Reis, Ana L. T. G. Ruiz, Paulo M. Imamura, João E. de Carvalho, Giovanna B. Longato

**Affiliations:** 1Research Laboratory in Molecular Pharmacology of Bioactive Compounds. São Francisco University, Bragança Paulista 12916–900, SP, Brazil; giovanna.longato@usf.edu.br; 2Posgraduate program in Health Science, São Francisco University, Bragança Paulista 12916–900, SP, Brazil; 3Chemical, Biological and Agricultural Pluridisciplinary Research Center (CPQBA), University of Campinas–UNICAMP, Paulínia 13148–218, SP, Brazil; yumiokuboh@gmail.com (M.Y.O.); adriana_biotec@yahoo.com.br (A.D.T.); paula.p21@gmail.com (P.P.P.); 4Posgraduate program in dentistry, Piracicaba Dental School, University of Campinas, Piracicaba 13 414–903, SP, Brazil; ana.ruiz@fcf.unicamp.br (A.L.T.G.R.); carvalho@fcf.unicamp.br (J.E.d.C.); 5Molecular Oncology Research Center, Barretos Cancer Hospital, Barretos 14.784–400, SP, Brazil; nr.marcela2@gmail.com (M.N.R.); vivianeaos@gmail.com (V.A.O.S.); ruireis.hcb@gmail.com (R.M.R.); 6Life and Health Sciences Research Institute (ICVS), School of Medicine, University of Minho, 4710–057 Braga, Portugal; 7ICVS/3B’s–PT Government Associate Laboratory, 4710–057 Braga, Portugal; 8Faculty of Pharmaceutical Sciences, University of Campinas, UNICAMP, Campinas 13081–970, SP, Brazil; 9Institute of Chemistry, University of Campinas–UNICAMP, P.O. Box 6154, Campinas 13083–970, SP, Brazil; imam@iqm.unicamp.br

**Keywords:** coronarin D, glioblastoma, apoptosis, cell cycle arrest, natural products

## Abstract

Glioblastoma (GBM) is the most frequent and highest–grade brain tumor in adults. The prognosis is still poor despite the use of combined therapy involving maximal surgical resection, radiotherapy, and chemotherapy. The development of more efficient drugs without noticeable side effects is urgent. Coronarin D is a diterpene obtained from the rhizome extract of *Hedychium coronarium*, classified as a labdane with several biological activities, principally anticancer potential. The aim of the present study was to determine the anti–cancer properties of Coronarin D in the glioblastoma cell line and further elucidate the underlying molecular mechanisms. Coronarin D potently suppressed cell viability in glioblastoma U–251 cell line, and also induced G1 arrest by reducing p21 protein and histone H2AX phosphorylation, leading to DNA damage and apoptosis. Further studies showed that Coronarin D increased the production of reactive oxygen species, lead to mitochondrial membrane potential depolarization, and subsequently activated caspases and ERK phosphorylation, major mechanisms involved in apoptosis. To our knowledge, this is the first analysis referring to this compound on the glioma cell line. These findings highlight the antiproliferative activity of Coronarin D against glioblastoma cell line U–251 and provide a basis for further investigation on its antineoplastic activity on brain cancer.

## 1. Introduction

Comprising over 100 diseases, cancer is characterized by disordered cell growth and tissue invasion that can spread to other regions of the body, leading to metastasis. It is a multifactorial heterogeneous disease and one of the major worldwide causes of mortality [1]. Among all types of cancers, brain tumors are one of the less prevalent, accounting for about 2% of all types of malignant tumors, but considered one of the most worrying ones [2,3]. As the most frequent (70–75%), gliomas were originally characterized as lesions originated from glial cells, which play a role of support, protection, and nourishment for neurons in the central nervous system (CNS) [4]. However, stem–like cells within the CNS are now thought to be the cells of origin of several primary brain tumor types, including glioblastomas [5,6]. Gliomas are one of the most fatal tumors, presenting a high mortality rate, 29–35%, of the CNS tumors in adolescents and young adults [7,8]. Among gliomas, glioblastomas (GBM) are the most aggressive and frequent subtype [9,10]. GBM is highly invasive and presents a median survival of only 14.6 months, even after aggressive treatment with surgery, radiation, and chemotherapy. The difficulties in human GBM therapy are due to the pathological characteristic and numerous drug–resistance mechanisms. Temozolomide (TMZ) comprises the standard treatment for glioblastoma, but unlike classic chemotherapeutics, TMZ does not induce DNA damage or misalignment of segregating chromosomes directly. It is a DNA alkylating agent, which leads to base mismatches that initiate futile DNA repair cycles; eventually, DNA strands break, which in turn induces cell death. The addition of TMZ to the standard treatment protocol was hailed as a major breakthrough in GBM therapy. Despite this, patients’ prognosis remains dismal with a five–year overall survival below 10% [11]. For this reason, more efficient therapeutic approaches are required for enhancing the treatment effect [12].

One of these approaches is based on cell death induction. There are many cell death morphotypes described in the literature [13]. Searching for new anticancer agents, investigation into cell death mechanisms, such as apoptosis, necrosis, necroptosis or other under–explored forms of cell death, is a significant strategy to afford more selective and efficient drugs. Further, as many clinically–established drugs are based on natural products, there are still many researches focused on finding new compounds from natural products [14].

Native from Asia and the Pacific [15], *Hedychium coronarium* (Zingiberaceae family) is an invasive species in Brazil [16]. Popularly known as white garland–lily, butterfly lily, napoleon, narcissus, Olympia, white ginger or “lírio do brejo” (in Brazil), the *H. coronarium* rhizomes are used as a starch source [17] and in traditional medicine for treatment of inflammation, diabetes, and rheumatic pain, among other uses [15]. Among pharmacological evaluations, the ethanolic extract of *H. coronarium* rhizomes induces apoptosis on HeLa cells by promoting cell cycle arresting at G1 phase, upregulating p53, p21, and Bax expression as well as downregulating cyclin D1, cyclin–dependent kinases CDK–4, CDK–6, and Bcl–2 expression [18]. Moreover, chemical evaluation of these extracts afforded the isolation of several labdane–like diterpenes with anti–inflammatory action [19,20], antiallergic [21], antibacterial [22], and cytotoxic effects over A–549– lung cancer, SK–N–SH– human neuroblastoma, MCF–7 breast cancer, and HeLa cervical cancer cell lines [23].

One of these diterpenes, coronarin D, has been reported as a promising antiproliferative and anti–inflammatory agent. Coronarin D inhibits the β–hexosaminidase release in RBL–2H3 cells [21] in addition to increasing the in vivo inhibition of the acetic acid–induced vascular permeability in mice [19]. Further, Coronarin D exhibits antiproliferative, pro–apoptotic, anti–invasive, antiangiogenic, antiosteoclast, and anti–inflammatory activity by suppressing NF–κB and the gene products regulated by this pathway of osteoclastogenesis [24]. Recently, Coronarin D has been described as inducing apoptosis in human hepatocellular carcinoma (HCC) [25] and in human oral cancer (OSCC) [26] through the c–Jun N–terminal kinases (JNK) pathway while it has induced reactive oxygen species–mediated cell death in human nasopharyngeal cancer cells (NPC) through inhibition of p38 mitogen–activated protein kinase (MAPK) and activation of JNK [27].

Based on these significant activities, the present study sought to further elucidate the Coronarin D mechanism of action on cell death of the human tumor cell line U–251 (glioblastoma). As far as we know, this is the first report concerning the Coronarin D mechanism of action on glioblastoma cancer cell line.

## 2. Results

### 2.1. Isolation and Characterization of Coronarin D

Coronarin D (Figure 1) was obtained from the dichloromethane crude extract of *Hedychium coronarium* rhizomes. The rhizomes were collected by Dr. Paulo Matsuo Imamura and identified at the herbarium of the State University of Campinas (UEC 163701). The identification of Coronarin D was done by comparison of experimental ^1^H– and ^13^C–NMR data (Appendix A) with those described by Itokawa et al. [28].

### 2.2. In Vitro Antiproliferative Activity Assay

Coronarin D presented an interesting antiproliferative activity (Figure 2, Table 1), with U–251 (glioblastoma), 786–0 (kidney), PC–3 (prostate), and OVCAR–3 (ovary) as the most sensitive ones, and total growth inhibition (TGI) values <50 µM.

The glioma cell line (U–251) was chosen to continue the in vitro experimental procedures, taking into account that the treatment for this tumor type is still scarce and requires alternative therapies, as previously mentioned. Considering the TGI value, the concentrations of 2.5, 5, and 10 μM were chosen to proceed with the cell cycle; concentrations of 10, 20, and 40 μM were chosen for the flow cytometry and 40 μM for the Western blot assay.

### 2.3. Cell Cycle Assay

Comparing to untreated U–251 cells, Coronarin D induced cell cycle arrest at G1 phase, in a concentration–dependent way and independent of time exposure (Figure 3). The increasing G1 subpopulation was concomitant with a significant reduction on cell subpopulations at S phase (Figure 3a) and G2/M phase (Figure 3a,b), proportionally to Coronarin D concentration. Figure 3c,d reveal the histogram of the most representative concentration (10 µM).

### 2.4. Phosphatidylserine (PS) Externalization Assay

According to Figure 4, after 12 h of U–251 exposition, the concentrations 20 and 40 μM reduced cell viability and increased the number of cells labeled with annexin V–PE (17.88% and 25.88%, consecutively) and doubly labeled with annexin V–PE/7–AAD (7.30 and 13.00%, consecutively). After 24 h of treatment with Coronarin D at 10, 20, and 40 μM, the cell viability was dramatically reduced in comparison with the control (63.4%, 52.00%, 28.88%) and there was an increase of cells labeled with annexin V–PE (26.32%, 23.18%, 22.75%) and doubly labeled with annexin V–PE/7–AAD (9.50%, 19.42%, 42.00%, consecutively). Coronarin D induces cell death through a concentration–dependent effect—the higher the concentration, the more advanced cell death process. The population of non–viable cells labeled only by 7–AAD did not increase significantly, indicating that the treatments with Coronarin D induced cell death characterized by phosphatidylserine exposure, being a type of programmed cell death.

### 2.5. Detection of Activated Caspases

The results obtained for caspases corroborate with annexin assay. The highest concentrations (20 and 40 μM) led to caspases activation without cell membrane disruption in 14.3% and 12.5% of cells, respectively. The percentage of cells doubly labeled, which means, caspases activation with cell membrane disruption increased for 20 μM concentration and this percentage was even higher for 40 μM concentration (49.7%) (Figure 5).

### 2.6. Mitochondrial Membrane Potential Assay

The induction of death by intrinsic apoptosis is usually triggered by some stimulus or stress that leads to a mitochondrial response and may result in the depolarization of its outer membrane. Untreated cells showed high intracellular fluorescence intensity indicating that mitochondria were able to sequester a greater amount of rhodamine 123, whereas in cells treated with Coronarin D at 20 μM and 40 μM for 6, 9, and 12 h, there was an intracellular fluorescence signal reduction (Figure 6a–c), being more intense at 12 h of treatment. This result suggests that Coronarin D induces loss of mitochondrial membrane preceding or concomitant with caspase activation and phosphatidylserine externalization.

### 2.7. Measurement of Hydrogen Peroxide (H_2_O_2_) Generation

The induction of death by intrinsic apoptosis is usually triggered by a stress that leads to depolarization of the outer membrane of mitochondria and the release of reactive oxygen species (ROS), more specifically hydrogen peroxide. This was measured by examining the fluorescence intensity of DCF. The intensity of fluorescence is proportional to intracellular hydrogen peroxide levels [30]. Data suggest that over 80% of the cell population presented high fluorescence intensity (DCF +) after 90 min of treatment with Coronarin D, even at the lowest concentration (10 μM), indicating the presence of H_2_O_2_ on these cells (Figure 7).

### 2.8. Western Blotting Assay

In order to confirm the cell signaling pathway of Coronarin D, some proteins involved with proliferation, cell death and cell cycle were evaluated after 24 h of treatment. Figure 8 revealed a decrease of total ERK protein, cleavage of poly (ADP–ribose) polymerases (PARP) and cleavage of caspases 3, 7, and 9 as well as increase of phosphorylation of ERK and p–H2AX histone. The p21 protein, related with cell cycle arrest, was also overexpressed. Of note, Coronarin D activated caspases 7 and 9, as well as PARP and p21 in a more intense way compared with the chemotherapeutic drug TMZ.

## 3. Discussion

Since Coronarin D has shown several biological activities as mentioned before and presents a potential clinical application in cancer therapy, it is important to have a clear understanding of its mechanism of action. In this study, we showed that most of the tumor cell lines evaluated were sensitive to the treatment with Coronarin D and, among them, U–251 (glioblastoma) was the cell line chosen to continue the evaluation of the mechanism of action of this compound. We demonstrated that Coronarin D induces cell cycle arrest at G1 phase and apoptosis of glioblastoma cells in a concentration–time dependent manner.

Many natural products can suppress proliferation by arresting cells at phases in the cell cycle [25]. The p21 is a small protein with 165 amino acids and belongs to the CIP/Kip family of CDK inhibitors. The p21 can arrest the cell cycle progression in G1/S and G2/M transitions by inhibiting CDK4,6/cyclin–D and CDK2/cyclin–E, respectively [31,32]. In addition, some studies have shown that H2AX is required for p21–induced cell cycle arrest after replication stalling [33]. The results herein presented indicate that Coronarin D inhibits glioblastoma (U–251) cell growth by inducing cell cycle arrest at G1 phase after increasing expression of p21, likely mediated by the phosphorylation of H2AX. 

Coronarin D was also able to trigger cell death with the activation of caspases 9, 3, and 7 and phosphatidylserine exposure, characteristics of apoptosis, and with further rupture of the cell membrane [13]. There was a gradual and time–dependent reduction of the mitochondrial membrane potential (MMP) in U–251 cells treated with Coronarin D, which is a feature of the intrinsic apoptotic pathway that occurs in response to various intracellular stress conditions centered on mitochondria [34]. The ROS production after the cell treatment with Coronarin D suggests that it can act as a second messenger, signaling to the activation of the apoptotic process, since this can activate effector caspases. In addition, ROS can lead to DNA damage that, in turn, activates the p21 pathway and results in cell cycle arrest [35].

Coronarin D led to an increase in the expression of protein kinase ERK, as well as PARP cleavage. ERK is part of the MAPK family and, when activated, can mediate mechanisms of cell proliferation and apoptosis [36,37]. Some studies have reported that the activation of ERK could be a result of DNA damage that subsequently leads to cell cycle arrest and apoptosis [38,39]. In addition, it is known that intracellular ROS lead to the activation of ERK and subsequent apoptosis [40,41,42,43]. Poly (ADP–ribose) polymerases (PARPs) are a family of enzymes involved in cellular homeostasis, including DNA transcription, cell–cycle regulation, and DNA repair [44]. This protein is really relevant in the apoptosis pathway, because it has a positive regulation in tumors and when it is inactivated leads to the cleavage of caspase 3 that is involved with the apoptosis process. Studies describe that some natural products are responsible for cleaving PARP as well as activating the caspase cascade as a mechanism of action on the induction of apoptosis [45,46]. Relating all these data reported we can suggest that Coronarin D could induce apoptosis involving ROS generation and ERK activation in U–251 cell line through an intrinsic and caspase–dependent pathway.

It has been reported in the literature that Coronarin D has pro–apoptotic potential, including potentiation of PARP cleavage and a reduction in the expression of anti–apoptotic gene products, such as apoptosis protein–1, TRAF–2 cellular inhibitory proteins, surviving, and Bcl–2 [24]. Consistent with our findings it has been shown that Coronarin D triggers apoptosis by activating caspase–dependent proteins and altering the expression of Bcl–2, Bcl–xL, and Bak in human hepatoma cell lines [25]. Moreover, Dimas et al. and Mahaira et al. demonstrated that compounds that contain labdane–type diterpenes triggered apoptosis in human colon cancer cells and myeloid leukemia cells [47,48].

The results obtained propose that Coronarin D elevates the generation of ROS (H_2_O_2_), which promotes phosphorylation of H2AX and consequently damage to the DNA. In addition, an increase in the expression of p21 leads cell cycle arrest at the check point between G1 and S. ROS generation also increases ERK phosphorylation and loss of mitochondrial membrane potential that allows the release of cytochrome c (not evaluated in this work), and consequently the cleavage of caspases (9, 3, and 7) and PARP protein. The caspase activation, in turn, leads cell to death through the mechanism of intrinsic apoptosis. In conclusion, as far as we know, this is the first study to elucidate the mechanism of action of Coronarin D in glioblastoma cell line and the results obtained highlight Coronarin D as a promising anticancer compound in Figure 9.

## 4. Materials and Methods

### 4.1. Chemicals and Equipments

Column chromatography on silica gel and thin layer chromatography (TLC) were obtained from Merck (Darmstad, Hesse, Germany). Culture medium RPMI 1640, fetal bovine serum, and Hank’s balanced salt solution (HBSS) were obtained from Gibco^®^ (Gaithersburg, MD, USA). Penicillin/streptomycin (1000 U/mL:1000 µg/mL) was purchased from LGC Biotechnology (Cotia, SP, BRA) and trypsin–EDTA 0.25% from Vitrocell^®^ (Campinas, SP, BRA). Sulforhodamine B (SRB), trizma base, trichloroacetic acid (TCA), DCFH–DA, and Rhodamine–123 dyes were obtained from Sigma–Aldrich^®^ (Allentown, PA, USA). Doxorubicin hydrochloride was obtained from Eurofarma (J urubatuba, SP, BRA). Temozolomide (TMZ) was acquired from Sigma–Aldrich. Dimethylsulfoxide (DMSO), silica gel, and dichloromethane were supplied by Merck (Darmstad, Hesse, Germany). Guava Cell Cycle^®^, Guava Nexin Reagent^®^, and Guava Multicaspase Kit^®^ were purchased from Millipore^®^ (Burlington, MA, USA). The nitrocellulose membranes were obtained from Hybond–C™Extra, Amersham Biosciences (Piscataway, NJ, USA). Anti–caspase 7, anti–p–H2–AX histone (Ser139), α–tubulin, anti–caspase 9, anti–p44/42 MAPK (ERK1/2), and anti–P21 were obtained from Cell Signaling and Bradford reagent was purchase from Bio–Rad Hybond–C™ (Hercules, CA, USA). The equipments used were rotary evaporator Buchi–Merck (Darmstad, Hesse, Germany), microplate reader (Molecular Devices^®^, model versaMax), and flow cytometer Guava EasyCyte Mini Flow Cytometry System (Millipore^®^).

### 4.2. Isolation of Coronarin D Obtained from Hedychium Coronarium 

The rhizomes of *Hedychium coronarium* were collected in January 2011 in the city of Embu, São Paulo, Brazil. Dried–milled rhizomes (480 g) were extracted with dichloromethane (4 L) by static maceration (7 days) at room temperature followed by filtration. The plant residues were re–extracted twice more following the same procedure. After fluidic extract grouping and solvent evaporation under a vacuum, the final crude extract (32.5 g) was purified by column chromatography on silica gel using hexane and hexane/ethyl acetate mixtures (1%, 2%, 5%, 10%, 20%, 50%) as eluent. Coronarin D was isolated after successive column chromatography from fractions obtained by 20% and 50% eluents. The complete identification of Coronarin D was done through experimental spectral data (^1^H– and ^13^C–NMR) in comparison to those described in the literature [28].

### 4.3. In Vitro Anticancer Activity Assay

#### 4.3.1. Cell Culture

The tumor cell lines U–251 (glioblastoma), MCF7 (breast), NCI/ADR–RES (resistant ovary), 786–0 (kidney), NCI–H460 (lung), PC–3 (prostate), OVCAR–3 (ovary), HT–29 (colon), and K–562 (leukemia) were kindly provided from the National Cancer Institute at Frederick, Maryland, USA. The non–tumor cell line HaCaT (keratinocytes) was kindly provided by FOP/Unicamp. Cells were cultured in complete medium (RPMI–1640) supplemented with 5% heat fetal bovine serum and 1% penicillin/streptomycin at 37 °C with 5% CO_2_.

#### 4.3.2. Antiproliferative Activity

Antiproliferative activity was assessed by the sulforhodamine B (SRB) colorimetric assay as previously reported [30]. First, the stock solution of Coronarin D (0.3 mM) was prepared aseptically using DMSO followed by serial dilution in complete medium. The cells were seeded in 96–well plates (3 × 10^4^ cells/mL, 100 μL/well), incubated for 24 h and treated with Coronarin D at final concentrations of 0.79, 7.9, 78.5, and 785.0 μM (100 μL/well), in triplicate, and then incubated for 48 h at 37 °C in 5% CO_2_. A second plate, named T0, was prepared to infer the absorbance value of untreated cells at the sample addition moment. The antineoplastic agent doxorubicin hydrochloride was used as a positive control, at final concentrations of 0.0431, 0.431, 4.31, and 43.1 μM (100 μL/well), in triplicate. The final concentration of DMSO (≤0.25%) did not affect cell viability [46]. Subsequently, the cells were fixed with 50% trichloroacetic acid and stained with SRB protein dye (0.4%). Determination of protein content was performed using a microplate reader (Molecular Devices^®^, VersaMax model) at 540 nm. Using the absorbance values, the cell growth (%) for each cell line, at each sample concentration, was calculated considering at 100% of cell growth the difference between the absorbances of untreated cells after 48 h incubation (T_1_) and at the sample addition moment (T_0_). The curve cell growth vs. sample concentration was plotted and the effective concentration TGI (concentration required for total cell growth inhibition) was calculated by sigmoidal regression using the Origin 8.0 software (OriginLab Corporation, Northampton, MA, USA).

### 4.4. Cell Cycle Analysis

This experiment was done using Guava^®^ Cell Cycle reagent, following the fabricant’s instructions. Briefly, U–251 cells (3 × 10^4^ cells/well) in 12–well plates with complete medium were incubated at 37 °C with 5% CO_2_ for 24 h. Then, the complete medium was replaced by RPMI medium without fetal bovine serum (FBS) for cell cycle synchronization and cells were incubated for another 24 h. After that, the cells were treated with Coronarin D (2.5, 5, and 10 μM) during the 24 h. The cells were harvested, collected, centrifuged (5 min, 2500 rpm), and the supernatant discarded. After fixation with 70% cold ethanol (24 h, 4 °C), each cell suspention was centrifuged and washed with PBS, the supernatant was discarded, and then the Guava Cell Cycle reagent (200 μL/cell suspension) was added. After 20 min at room temperature in the dark, each cell suspension was analyzed (5000 events/replicate) by flow cytometry. Using the Guava Cell cycle^®^ software (Austin, TX, USA), the subpopulations at G1, S, and G2/M phases of the cell cycle were quantified in percentage. The analyses were done in biological triplicate of two experiments. Statistical evaluation was done by two–way ANOVA followed by Bonferroni test using GraphPad Prism.

### 4.5. Phosphatidylserine (PS) Externalization Assay

This experiment was done using Guava^®^ Nexin Reagent, following the fabricant’s instructions. Briefly, U–251 cells (3 × 10^4^ cells/well) in 12–well plates with complete medium were incubated at 37 °C with 5% CO_2_ for 24 h. Then cells were treated with Coronarin D (10, 20, and 40 μM) for 12 and 24 h. After trypsinization and washing, each cell suspension was stained with 100 μL of Guava Nexin Reagent that consists of annexin–V conjugated with phycoeritrin (PE) and 7–aminoactinomycin D (7–AAD) for 20 min at room temperature in the dark and then analyzed (2000 events/replicate) by flow cytometry. Using the Guava Nexin^®^ software (Austin, TX, USA), each cell population was registered and quantified at four cell subpopulations named as viable cells ((–) annexin–V/PE (–) 7–AAD); only PS externalization ((+) annexin–V/PE (–) 7–AAD); both PS externalization and membrane permeabilization ((+) annexin–V/PE (+) 7–AAD); only membrane permeabilization ((–) annexin–V/PE (+) 7–AAD). The analyses were done in biological triplicate of two experiments. Statistical evaluation was done by two–way ANOVA followed by Bonferroni test using GraphPad Prism software.

### 4.6. Detection of Activated Multicaspases

This experiment was done using Guava^®^ Caspase Kit, following the fabricant’s instructions. Briefly, U–251 cells (3 × 10^4^ cells/well) in 12–well plates with complete medium were incubated at 37 °C with 5% CO_2_ for 24 h. Then cells were treated with Coronarin D (10, 20, and 40 μM) for 12 and 24 h. After that, the caspase inhibitor probe (10 μL/well) was added. It comprises a sulforhodamine derivative of valylalanylaspartic acid fluoromethyl ketone (SR–VAD–FMK). After incubation (1 h in the dark) and medium discarding, the cells were washed, trypsinized, and centrifuged. Each resulting cell pellet was resuspended in wash buffer (100 μL/pellet) and co–stained with the caspase–7–AAD working solution (200 μL/pellet) After 10 min at room temperature in the dark, cells were analyzed (2000 events/replicate) by flow cytometry. Using the Guava^®^ MultiCaspase software (Austin, TX, USA), each cell population was registered and quantified at four cell subpopulations named as viable cells (SR–VAD–FMK (–) and 7–AAD (–)); only activated caspases (SR–VAD–FMK (+) and 7–AAD (–)); both activated caspases and membrane permeabilization (SR–VAD–FMK (+) and 7–AAD (+)) and only membrane permeabilization (SR–VAD–FMK (–) and 7–AAD (+)). The analyses were done in biological triplicate of two experiments. Statistical evaluation was done by Two–way ANOVA followed by Bonferroni test using GraphPad Prism software.

### 4.7. Mitochondrial Membrane Potential Assay

Briefly, U–251 cells (3 × 10^4^ cells/well) in 12–well plates with complete medium were incubated at 37 °C with 5% CO_2_ for 24 h. Then cells were treated with Coronarin D (20 and 40 µM) for 6 h. Rhodamine–123 solution (1 μg/mL in medium RPMI 1640 plus 10% FBS, 1 mL/well) was added after medium remoting. After 15 min, the medium was aspirated and all cells were washed with RPMI 1640 plus 10% FBS (1 mL/well, twice), trypsinized, and collected. The cells were analyzed (5000 events/replicate) by flow cytometry. Using the Guava^®^ Express Pro software (Austin, TX, USA) each cell population was registered and quantified at two cell subpopulations named as viable cells (Rhodamine–123(+)); altered mitochondrial membrane potential (Rhodamine–123(–)). The analyses were done in biological duplicate of one experiment. Statistical evaluation was done by one–way ANOVA followed by Tukey’s test using GraphPad Prism software.

### 4.8. Measurement of Hydrogen Peroxide (H_2_O_2_) Generation

DCFH–DA (Dichlorodihydro–fluorescein diacetate) is a stable, fluorogenic, and non–polar compound, which can readily diffuse into cells and get deacetylated by intracellular esterases to a non–fluorescent 2,7–dichlorodihydrofluorescein (DCFH) which is later oxidized by intracellular hydrogen peroxide into highly fluorescent 2,7–dichlorofluorescein (DCF). The intensity of fluorescence is proportional to intracellular hydrogen peroxide levels [30].

Briefly, U–251 cells (3 × 10^4^ cells/well) in 12–well plates with complete medium were incubated at 37 °C with 5% CO_2_ for 24 h. After medium removing, the cells were washed with Hank’s buffered salt solution (HBSS, 1 mL/well), treated with DCFH–DA solution (10 μM in HBSS, 1 mL/well), followed by 30 min incubation in the dark. After probe removing and cell washing with HBSS buffer, cells were treated with Coronarin D (20 and 40 μM) for 90 min. After trypsinization, cell suspension in HSBB was analyzed (5000 events/replicate) by flow cytometry. Using the Guava^®^ Express Pro software, each cell population was registered and quantified at two cell subpopulations named as viable cells (DCF(–)) and increased intracellular hydrogen peroxide level (DCF(+)). The analyses were done in biological duplicate of one experiment. Statistical evaluation was done by one–way ANOVA followed by Tukey’s test using GraphPad Prism software.

### 4.9. Western Blotting Assay

U–251 cells (1 × 10^6^ cells/well) in 6–well plates with complete medium were incubated at 37 °C with 5% CO_2_ for 24 h. Then cells were treated with a negative control (DMSO), Coronarin D (40 µM), and TMZ (used as positive control) for 24 h. TMZ was dissolved in DMSO to prepare a stock concentration of 1000 mM, which was further diluted in cell culture medium to working concentrations. The cells were exposed at 3 × IC_50_ concentration values of TMZ for 24 h in DMEM (0.5% FBS) [49]. After washing with PBS (1 mL/well, twice), cells were lysed with lysis buffer (70 µL/well); Tris 50 mM pH 7.6–8, NaCl 150 mM, EDTA 5 mM, Na_3_VO_4_ 1 mM, NaF 10 mM, Na pyrophosphate 10 mM, and 1% NP–40 and supplemented with a cocktail of inhibitors (DTT– Dithiothreitol, leupeptin hemisulfate, aprotinin, PSMF– Phenylmethylsulfonyl fluoride, and EDTA) for 1 h, followed by centrifugation (4 °C, 15 min, 13,000 rpm). The protein concentration was determined by Bradford reagent [50]. All samples (untreated and Coronarin d–treated cells) were total cellular protein (20 μg protein/sample) and separated by electrophoresis on 10% gradient gels in SDS–PAGE and blotted onto nitrocellulose membranes by electroblotting in transfer buffer (Trizma base, glycine, distilled water, and methanol). Then the membranes were blocked with skimmed milk powder solution at 5% diluted in tris–buffered saline and Tween 20 (TBST) and incubated overnight with the primary antibodies diluted in bovine serum albumin (BSA) solution (5% in TBST). The primary antibodies were anti–caspase 7, anti–cleaved caspase 3, anti–caspase 9, anti–cleaved PARP, anti–P21 (dilution 1:1000), anti–p44/42 MAPK (ERK1/2), and anti–p–H2–AX histone (Ser139) (dilution 1:500). The antibody α–tubulin was used as the control. After this, the membranes were incubated with peroxidase–conjugated secondary antibody anti–mouse or anti–rabbit (1:5000) and also diluted in BSA solution (5% in TBST). The proteins levels were detected using the enhanced chemiluminescence (ECL–GE) method. The detection of the chemiluminescent signal was performed in the Photo Quant LAS 4000 mini (GE) photo documentation system and the bands were analyzed and quantified using Image J software (obtained from imagej.nih.gov/ij/download/).

### 4.10. Statistical Analysis 

The data are provided as the final result with mean ± standard error. For the statistical analysis of the experiments two–way analysis of variance (ANOVA) was used. All analyses were performed with significance level at *p* < 0.05, using GraphPad Prism version 8.0.0 for Windows, GraphPad Software (San Diego, CA, USA).

## Figures and Tables

**Figure 1 molecules-24-04498-f001:**
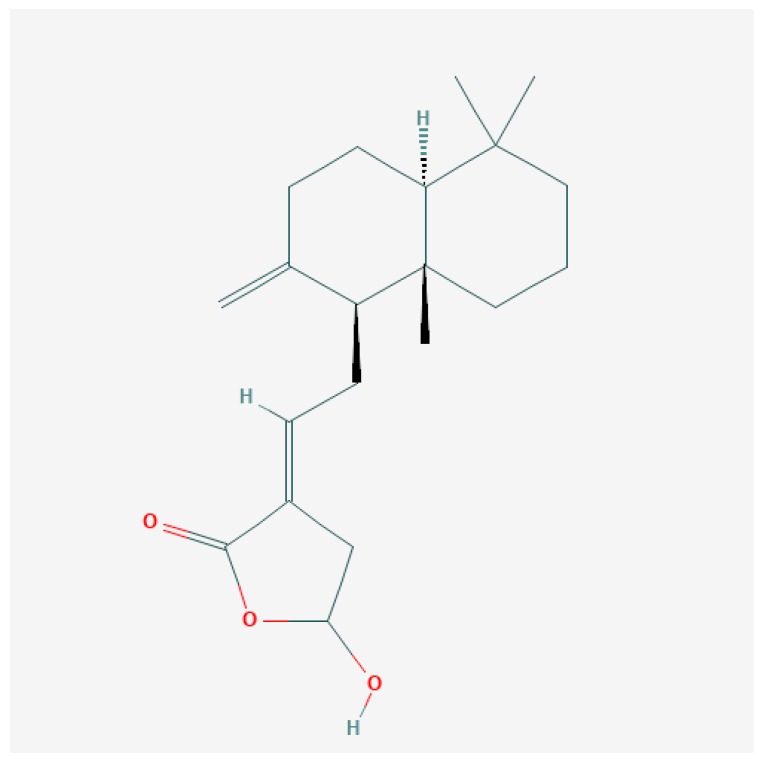
Coronarin D molecular structure, data from [29].

**Figure 2 molecules-24-04498-f002:**
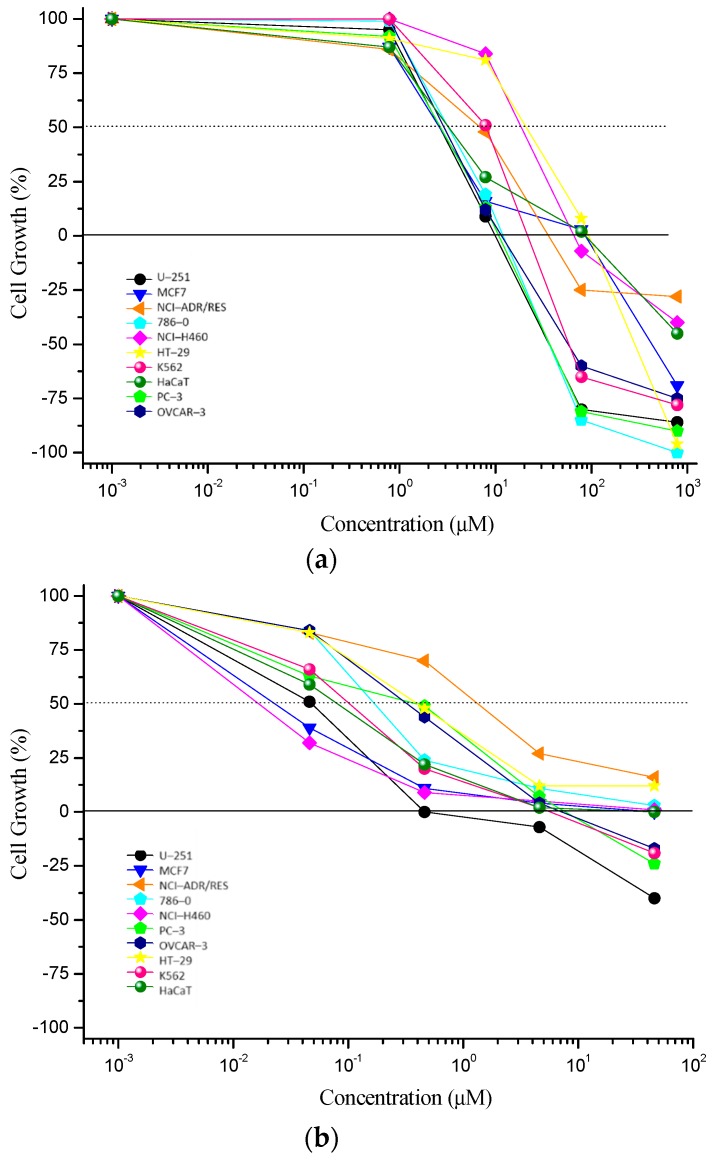
In vitro antiproliferative activity of (**a**) Coronarin D and (**b**) doxorubicin hydrochloride (positive control) after 48 h of treatment. Concentration range: 0.785–785 µM for Coronarin D; 0.043–43.1 µM for doxorubicin hydrochloride. Human tumor cell lines: U–251 (glioblastoma), MCF7 (breast), NCI–ADR/RES (multidrug resistant ovary), 786–0 (kidney), NCI–H460 (lung, non–small cells tumor), PC–3 (prostate), OVCAR–3 (ovary), HT–29 (colon), K562 (chronic myelogenous leukemia). Human non–tumor cell line: HaCaT (keratinocyte).

**Figure 3 molecules-24-04498-f003:**
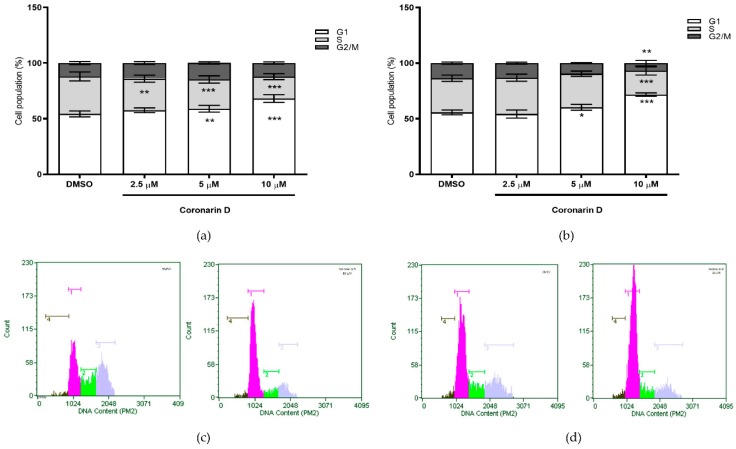
Quantification of U–251 in phases G1, S, and G2 after (**a**) 24 h and (**b**) 48 h of treatment with vehicle (DMSO) and Coronarin D at concentrations of 2.5, 5, and 10 μM. Histograms of the most representative concentration (10 μM) are presented at (**c**) 24 h and (**d**) 48 h. The values were expressed as mean ± standard deviation of two replicates of the same experiment. * *p* < 0.05; ** *p* < 0.01 and *** *p* < 0.001. (Two–way ANOVA: Bonferroni).

**Figure 4 molecules-24-04498-f004:**
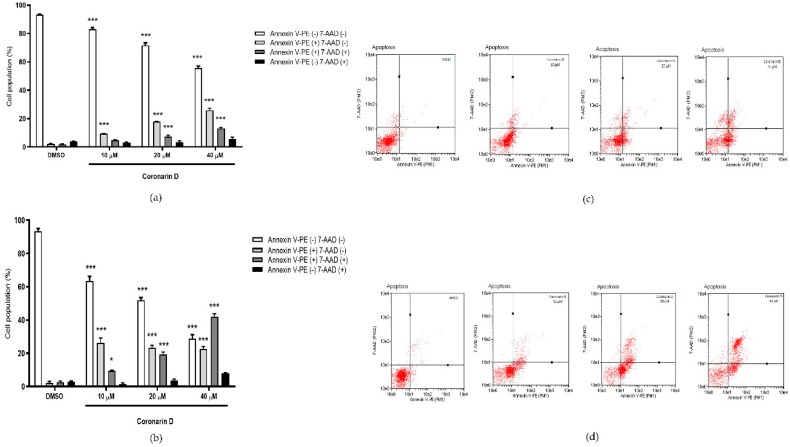
Percentage of U–251 cells stained with annexin V–PE and 7–AAD after (**a**) 12 h and (**b**) 24 h of treatment with vehicle (DMSO) and Coronarin D at 10, 20, and 40 μM concentrations. Dotplots are presented at (**c**) 12 h and (**d**) 24 h. The values are expressed as mean ± standard deviation of two replicates of the same experiment. * *p* < 0.05 and *** *p* < 0.001. (Two–way ANOVA: Bonferroni).

**Figure 5 molecules-24-04498-f005:**
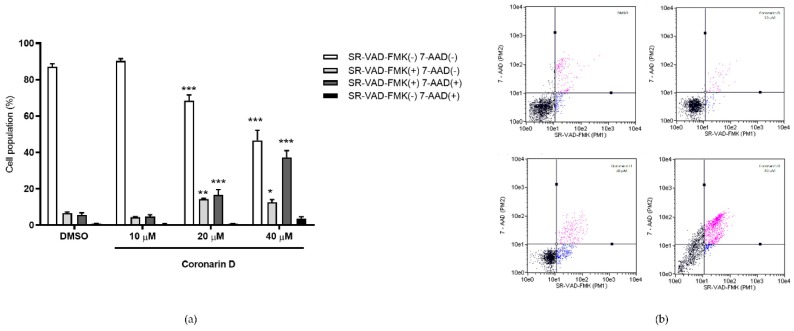
(**a**) Percentage of U–251 cells stained with SR–VAD–FMK and 7–AAD after 24 h of treatment with vehicle (DMSO) and Coronarin D at 10, 20, and 40 μM concentrations. Dotplots are presented at (**b**). The values are expressed as mean ± standard deviation of two replicates of the same experiment. * *p* < 0.05, ** *p* < 0.01, and *** *p* < 0.001. (Two–way ANOVA: Bonferroni).

**Figure 6 molecules-24-04498-f006:**
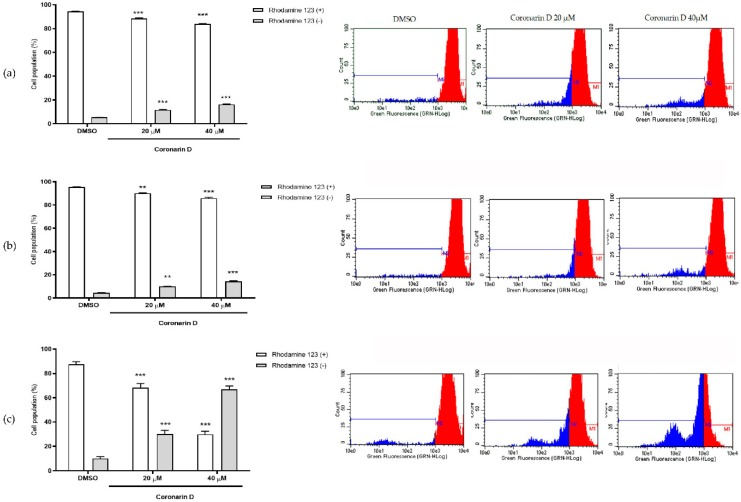
Percentage of cells with high (rhodamine +) and low (rhodamine –) intracellular fluorescence intensity after (**a**) 6 h, (**b**) 9 h, and (**c**) 12 h of treatment with vehicle (DMSO) and Coronarin D at 20 μM and 40 μM. The values were expressed as mean ± standard deviation of two replicates of the same experiment. ** *p* < 0.01 and *** *p* < 0.001. (ANOVA Two–way: Bonferroni).

**Figure 7 molecules-24-04498-f007:**
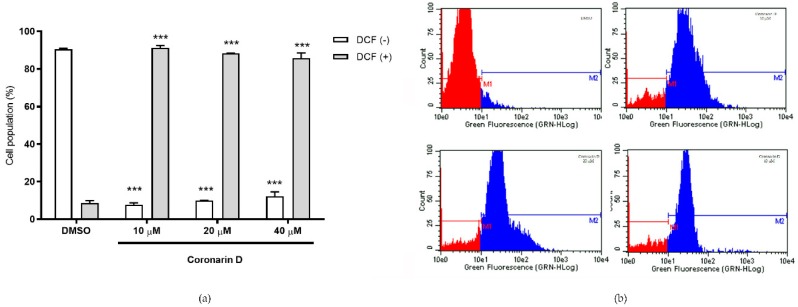
(**a**) Percentage of cells with high (DCF +) and low (DCF –) 2,7–dichlorofluorescein intracellular fluorescence intensity after 90 min of treatment with vehicle (DMSO) and Coronarin D at 10, 20, and 40 μM. Histograms are presented at (**b**). The values are expressed as mean ± standard deviation of two replicates of the same experiment. *** *p* < 0.001. (ANOVA Two–way: Bonferroni).

**Figure 8 molecules-24-04498-f008:**
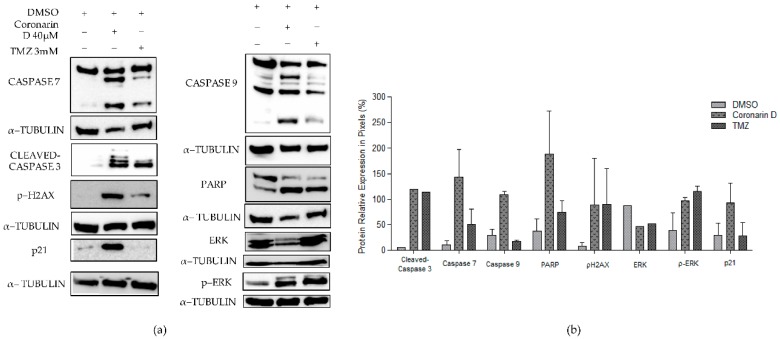
Effects of Coronarin D and temozolomide (TMZ) on expression of caspase 9, caspase 7, cleaved–caspase 3, poly (ADP–ribose) polymerase (PARP), p–H2AX, ERK, and p–ERK proteins in glioblastoma cell line (U–251). The α–tubulin was used as a positive control. The values for caspase 7, caspase 9, PARP, p–H2AX, p–ERK, and p21 were expressed as mean ± standard deviation of three replicates of the same experiment. Values for cleaved–caspase 3 and ERK were obtained from one experiment.

**Figure 9 molecules-24-04498-f009:**
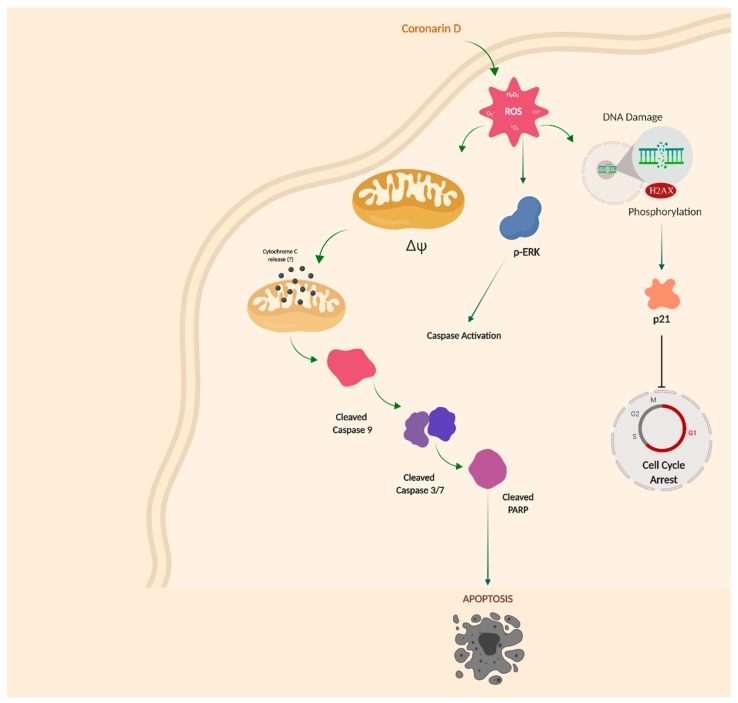
Overview of proposed mechanism of action by Coronarin D in U–251 glioblastoma cell line.

**Table 1 molecules-24-04498-t001:** Antiproliferative effect of Coronarin D and doxorubicin hydrochloride expressed as the concentration required for total growth inhibition (TGI, µM) after 48 h of exposition.

Cell Lines	Coronarin D	Doxorubicin Hydrochloride
	TGI (μM)	TGI (μM)
U–251	18.6 ± 0.5	2.4 ± 0.4
MCF7	105.0 ± 5.1	11.6 ± 1.8
NCI–ADR/RES	550.1 ± 79.9	>43.1 *
786–0	36.4 ± 4.2	17.8 ± 3.4
NCI–H460	640.8 ± 11.3	26.7 ± 1.8
PC–3	17.1 ± 0.6	21.4 ± 4.2
OVCAR–3	41.6 ± 2.1	19.2 ± 1.9
HT–29	534.1 ± 31.8	>43.1 *
K562	56.6 ± 2.4	10.0 ± 1.2
HaCaT	12.9 ± 1.7	1.1 ± 0.1

TGI (concentration required for total growth inhibition of each cell line) values expressed as mean ± standard error of two independent experiments. *: TGI values higher than the highest experimental concentration. Human tumor cell lines: U–251 (glioblastoma), MCF7 (breast), NCI–ADR/RES (multidrug resistant ovary), 786–0 (kidney), NCI–H460 (lung, non–small cells tumor), PC–3 (prostate), OVCAR­–3 (ovary), HT–29 (colon), K562 (chronic myelogenous leukemia). Human non–tumor cell line: HaCaT (keratinocyte).

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
