# Peer review of "Coronarin D Induces Apoptotic Cell Death and Cell Cycle Arrest in Human Glioblastoma Cell Line"

_molecules, 2019, doi:10.3390/molecules24244498_

Round 1
Reviewer 1 Report
The manuscript of Franco et al. deals with the investigation of the antiproliferative activity of Coronarin D on the human glioblastoma cell line U-251. In particular, the authors aim to elucidate the mechanism of action on cell death induced by the molecule.
In my opinion, the paper is interesting and deserve the publication on Molecules after addressing some concerns:
The authors only hypothesize that Coronarin D induces DNA damage effect. In order to support this conclusion, they should evaluate more in-depth this effect by performing more-addressed assays that use the specific γH2AX antibody or the Comet assay. Move the figure S1 (concerning the chemical structure of the Coronarin D) in the text. Specify what the acronym DCF stands for. Report the 1H and 13C NMR spectra in the supporting material.
Author Response
The authors only hypothesize that Coronarin D induces DNA damage effect. To support this conclusion, they should evaluate more in-depth this effect by performing more-addressed assays that use the specific γH2AX antibody or the Comet assay.
We agree with the reviewer that it should be interesting to evaluate more in-depth this effect by performing more-addressed assays to evidence damage effect. We used a specific H2AX phosphorylated antibody (Phospho-Histone H2AX (Ser139) as presented in figure 7 (now named Figure 8) and we observed an intense band, very different from DMSO control. This result allied to ROS elevation and cell cycle arrest through p21 mechanisms, highlight the effect of Coronarin D in damaging DNA. We are afraid we would not be able to perform a more-addressed assay at a reasonable term, so, respectfully, we ask the reviewers to reconsider this question.
Move the figure S1 (concerning the chemical structure of the Coronarin D) in the text.
Figure S1 was moved in the text, named as Figure 1.
Specify what the acronym DCF stands for.
Now it is specified at 2.7 item
Report the 1H and 13C NMR spectra in the supporting material.
The 1H and 13C NMR spectra were inserted in the supporting material as Figures S1 and S2.
Reviewer 2 Report
Comments:
Coronarin D was isolated from Hedychium coronarium Did the plant was correctly identified and confirmed by a local Taxonomist? Does dry specimen of the plant stored in the herbarium? Figure 1, y-axis should be labelled as “cell viability” For all the assays done with flow cytometry, provide histogram for each treatment group to support the graph. Figure 4 and 6 was not fully provided. Provide statistical analysis for the graph in figure 7. Also, the protein expression levels should be in either fold or in relative percentage (100%). For the phosphorylated (active) proteins (p21, H2AX and ERK), their total unphosphorylated form should be used as base control, but in this report, authors used common internal control (α-tubulin). Also, ERK should have 2 forms (ERK1/2). In all experiments at least 2 concentrations (20 and 40 µM) were used except western blot assay, why? In Materials and Methods section, both the concentrations are mentioned, but it was missing in the figure 7. Authors should show the protein expression pattern at 20 µM concentration. For ROS assay by DCF-DA, use H2O2 as positive control. In figure 7, TMZ (Temozolomide) was used at concentration of 3 mM. 3 mM TMZ was too high. On what basis TMZ at this concentration was chosen? Details of the TMZ was neither mentioned in Materials and Methods section, nor discussed in discussion section.
Author Response
Coronarin D was isolated from Hedychium coronariumDid the plant was correctly identified and confirmed by a local Taxonomist? Does dry specimen of the plant stored in the herbarium?
The rhizomes were identified at the herbarium of State University of Campinas (UEC 163701). This information was included in 2.1 section.
Figure 1, y-axis should be labeled as “cell viability”
Respectfully, the correct label for the y-axis is cell growth, once we have negative values. Cell viability is for a range between 0-100 %. As we consider a T0 plate, we use the following formula:
Percentage growth= 100×[(T-T0)/(C-T0)]
If T is greater than or equal to T0, and if T is less than T0, we consider percentage growth = 100×[(T-T0)/T0)].
T is the optical density of test,
C is the optical density of control,
T0 is the optical density at time zero.
From the percentage growth, a dose-response curve was generated and GI50 values were interpolated.
For all the assays done with flow cytometry, provide histogram for each treatment group to support the graph. Figure 4 and 6 was not fully provided.
All the histograms and dotplots were provided.
Provide statistical analysis for the graph in figure 7. Also, the protein expression levels should be in either fold or relative percentage (100%). For the phosphorylated (active) proteins (p21, H2AX, and ERK), their total unphosphorylated form should be used as base control, but in this report, the authors used common internal control (α-tubulin). Also, ERK should have 2 forms (ERK1/2).
We agree with the reviewer`s suggestion and improved it in accordance with the revised Figure 7 (now named Figure 8). We also agree with the reviewer`s comment that we should normalize the phosphorylated (active) proteins (H2AX and p21) with total form. However, in our laboratory, we do not have these antibodies available. Moreover, in numerous studies have used house-keeping gene proteins like actin/tubulin as base control in the absence of those. Regarding Erk1/2 we provided now a figure that reveals the two forms (ERK1/2) we also included total ERK protein performed in uniplicate experiment.
In all experiments at least 2 concentrations (20 and 40 µM) were used except western blot assay, why? In the Materials and Methods section, both the concentrations are mentioned, but it was missing in figure 7. The authors should show the protein expression pattern at 20 µM concentration.
As 40uM was the concentration that presented a major effect, we did all the western blot assays only with it. We have corrected it in the materials and methods section. We are sorry for this mistake.
For ROS assay by DCF-DA, use H2O2 as positive control.
We agree with the reviewer that a positive control might be necessary, but many articles do not present it for this test. Considering the expressive intensity of cell population labeled with DCF and that this was not an isolated assay (once the results corroborate with those obtained from membrane depolarization and caspases activation), respectfully, we ask the reviewer to accept it for publication on this way. We are sorry for it, but we are afraid we would not be able to replicate this test at a reasonable term.
In figure 7, TMZ (Temozolomide) was used at a concentration of 3 mM. 3 mM TMZ was too high. On what basis TMZ at this concentration was chosen?
We also agree with the reviewer`s comment about Temozolomide (TMZ) concentration. TMZ chemotherapy has been used in our study as a positive control of cell death. Since the effect of TMZ occurs after 72h, we performed the experiment with a concentration close to 3X IC50 values of 72 h [1], a high dose, but enough to promote its apoptosis-inducing effect at 24h as shown in figure 7 (now named 8).
Details of the TMZ was neither mentioned in the Materials and Methods section nor discussed in the discussion section
Details of TMZ were provided at Introduction, Materials and Methods and Results as indicated with Track change.
[1] Ho Yang S., Li S., Lu G., Xue H., Kim D. H., Zhu J., Liu Y. Metformin treatment reduces temozolomide resistance of glioblastoma cells. Oncotarget. 2016; 7: 78787-78803.
We hope the revised manuscript will better suit the Journal Molecules.
Sincerely,
Yollanda E. M. Franco
Round 2
Reviewer 1 Report
Manuscript was sufficiently improved after revisions and can be now be published in the present form.
Reviewer 2 Report
Authors have carried out the corrections and suggestions raised. Manuscript can be accepted for the publication.